# Application of the Homologous Modeling Technique for Precision Medicine in the Field of Oral and Maxillofacial Surgery

**DOI:** 10.3390/jpm12111831

**Published:** 2022-11-03

**Authors:** Hiroyuki Nakano, Kei Suzuki, Kazuya Inoue, Yoichiro Nakajima, Katsuaki Mishima, Takaaki Ueno, Noboru Demura

**Affiliations:** 1Department of Oral and Maxillofacial Surgery, Kanazawa Medical University, Kahoku 920-0293, Ishikawa, Japan; 2Department of Oral Surgery, Osaka Medical and Pharmaceutical University, Takatsuki 569-8686, Osaka, Japan; 3Department of Oral and Maxillofacial Surgery, Yamaguchi University, Ube 755-8505, Yamaguchi, Japan; 4Orthodontic Division of Oral-Maxillofacial Surgery, Kanazawa Medical University, Kahoku 920-0293, Ishikawa, Japan

**Keywords:** precision medicine, mandible, reconstruction, homologous modeling technique, oral and maxillofacial surgery

## Abstract

In the field of oral and maxillofacial surgery, establishment of a new method for predicting morphology is desirable. Therefore, the purpose of the present study was to establish a new method for predicting the original shape of a mandibular defect site using the homologous modeling technique. This study used data from 44 patients who underwent computed tomography in the Department of Oral Surgery at Osaka Medical College. Two types of homologous models were constructed: total mandible (TM) and half mandible (HM). Principal component analysis (PCA) was performed using point cloud data of the homologous model M and homologous model HM, and a multiple regression equation was created using the PC value of TM as the object variable and PC value of HM as the explanatory variable. The predicted PC (M) was created from PC (HM) using a regression formula, back-calculated from point cloud data from PC (M), to create the predicted mandible model. Finally, the original image (TC-M) and estimated mandible were superposed and examined. The mean absolute error between the predicted mandible and actual mandible was 1.04 ± 1.35 mm. We believe that this method will be applicable in actual clinical practice.

## 1. Introduction

In the field of oral and maxillofacial surgery, segmental mandibular defects caused by benign or malignant diseases usually result in compromise of facial esthetics and malfunction of mastication, respiration, phonation, and deglutition, which might severely impair the quality of life of patients. The management of mandibular defects is mandatory for surgeons, not only to rebuild the original contour of the face but also to rehabilitate the relative functions. The development of virtual surgical simulation of lesion resection, flap cutting, and positioning to the defect site before stereolithographic model fabrication has greatly increased the predictability of surgery [1]. Recently, efforts have been made by engineers and surgeons to design surgical templates for accurately carrying out the surgery according to the simulation [2,3]. On the other hand, when events such as trauma and neoplasia destroy the mandible, the original shape may need to be predicted for treatment. In such cases, the mandible is reconstructed conventionally, predicting the shape of the fractured part based on the shape of the opposite side. However, a completely symmetrical mandible is extremely rare and lateral differences in mandible morphology are normal. Therefore, establishment of a new method for predicting morphology is desirable.

The homologous modeling technique is an image analysis technique that involves pasting common template data, in which the number of polygons is determined for 3-dimensional (3D) data, while matching a set of anatomical landmarks. Originally, the human body data measured by the 3D shape scanner were just a set of data points and did not contain information of the human body. To enable statistical processing of the 3D shape of the human body, the data for any individual should be reconstructed to consist of the same number of data points with the same topology. This is called modeling. In particular, when we define data points such that each data point has the same anatomical meaning, we call it homology modeling. Through homologous modeling, anatomical information is added to the shape data of the human body and reconstructed into data with high utility value. This technique enables statistical processing, such as multivariate analysis, by creating a homologous model in which the number of polygons of the 3D data is matched and the point group data for the vertices of each polygon are used. In addition, Mochimaru et al. [4,5,6,7] advanced this method in developing a technique for reconstructing images by back-calculation from the point cloud data. A basic modeling technology based on target surface subdivisions has been developed and applied to foot and body contours. A homologous model can be used to represent the human body model in terms of data with the same number of points and topology. In addition, the reference body model (template model) is automatically deformed to correspond with other body model using the free-form deformation method.

Therefore, the purpose of the present study was to establish a new method for predicting the original shape of a mandibular defect site using the homologous modeling technique.

## 2. Materials and Methods

All experimental protocols were approved by the institutional ethics committee at Osaka Medical and Pharmaceutical University, Osaka, Japan. Informed consent was obtained from all patients.

### 2.1. Data Acquisition

CT data from patients with trauma or congenital or acquired disease resulting in jaw deformities were excluded from the study. In addition, CT data from patients with less than 14 remaining teeth were excluded, since the number of remaining teeth can affect the shape of the mandible.

Two types of homologous models were constructed: total mandible (TM) and half mandible (HM). Homologous modeling followed the methodology outlined by Suzuki et al. [8], as described in the next section (Figure 1).

### 2.2. Homologous Model

The CT data from patients were used to construct 3D images, and then the position of the mandible was determined based on the Frankfort horizontal plane. Landmarks were plotted on the surface of the 3D model (TM: 20 landmarks (Table 1); HM: 11 landmarks (Table 2)), using HBM-Rugle 3D-CT image measurement software (Medic Engineering, Kyoto, Japan) in stereolithographic format.

We generated template models of the mandible comprising approximately 8434 (TM) and 4179 (HM) polygons using Geomagic Studio 9 (Geomagic Inc., Morrisville, NC, USA). The template model automatically conformed to the individually scanned point cloud of the mandible by minimizing the external and internal energy functions. The external energy function was based on the Euclidean distance between data points on the template model and those on the patient’s database. The internal energy function was based on local deformation of the template model. Vertices of the template model were considered as anatomical landmarks with plotted landmarks, while vertices generated from surface subdivision conformed to the measured point cloud with minimum deformation of the initial template model. As described above, mandibles were constructed for each sample using Homologous Body Modeling software (HBM, Digita Human Technology, Tokyo, Japan) and HBM-Rugle software.

### 2.3. Statistical Analysis

Mandibles in the study were analyzed using principal component analysis (PCA). The number of parameters for the PCA were set as 16. All statistical analyses were performed using EZR (Saitama Medical Center, Jichi Medical University, Saitama, Japan), a graphical user interface for R (The R Foundation for Statistical Computing, Vienna, Austria). More precisely, EZR is a modified version of R Commander designed to add statistical functions frequently used in biostatistics.

PCA was performed using the point cloud data of homologous model M and homologous model HM, and a multiple regression equation was created using the principal component value of TM as the object variable and the principal component value of HM as the explanatory variable.
PC1(M) = aPC1(HM) + bPC2(HM) + cPC3(HM) + dPC4(HM) + z

The predicted PC (M) was created from PC (HM) using a regression formula, back-calculated form the point cloud data from PC (M), to create the predicted mandible model.

In addition, we created an image of the mandible by back-calculating the point cloud data from the estimated PC values. Finally, the original image (TC-M) and the estimated mandible were superposed and examined.

## 3. Results

This study used data from 44 patients (22 men, 22 women) who underwent computed tomography (CT) in the Osaka Medical and Pharmaceutical University Hospital between January 2018 and March 2019. The mean age of the patients was 41.3 ± 12.3 years (20–54 years old).

### 3.1. Total Mandible

The contribution of the most important PC was found to be 27.6% for the total mandible. The 16 PCs replicated more than 80% of the total variance, meaning that ≥80% of the mandible could be replicated using these 16 PCs (Table 3). Table 4 summarizes the interpretations of PCA components 1 to 3 (Figure 2).

### 3.2. Half Mandible

The contribution of the most important PC was found to be 27.9% for the total mandible. The 16 PCs replicated more than 85% of the total variance, meaning that ≥85% of the mandible could be replicated using these 16 PCs (Table 5).

### 3.3. Multiple Regression Equation

The regression equation was as follows:Estimated PCx(TM) = 0.11 + (−1.38) × PC1(HM) + (−0.07) × PC2(HM) + (−0.02) × PC3(HM) + 0.06 × PC4(HM) + 0.11 × PC5(HM) + (−0.04) × PC6(HM) + (−0.15) × PC7(HM) + (−0.12) × PC8(HM) + (−0.22) × PC9(HM) + (−0.03) × PC10(HM) + 0.16 × PC11(HM) + (−0.02) × PC12(HM) + (−0.17) × PC13(HM) + 0.30 × PC14PC(HM) + 0.20 × PC15(HM) + 0.50 × PC16(HM)

Table 6 show the estimated PCs.

In terms of face-to-face distance, the mean absolute error between the predicted mandible and actual mandible was 1.04 ± 1.35 mm (Figure 3).

This section may be divided by subheadings. It should provide a concise and precise description of the experimental results, their interpretation, as well as the experimental conclusions that can be drawn.

## 4. Discussion

In recent years, advances in scanner technology in the industrial field have also seen applications in the medical field. In particular, sizing surveys using 3D body scanners have been conducted in many studies since the end the twentieth century. The scanned data have been used to extract body dimensions, which require the surface shape to be described as a surface rather than a point cloud.

Matsumura et al. [9] explained the homologous model as follows:

“Template fitting is a method developed for this purpose in the field of computer graphics, in which the surface shape is described by a polygon mesh model. The first step in template fitting is the preparation of a mesh model that will serve as a template. Some vertices that constitute the template represent landmarks. The template is then deformed and fitted to the surface to minimize the distance between the template and point cloud while preserving the local shape features of the template as much as possible. The landmarks in the template match those in the point cloud. Using template fitting, all scan data can be assumed because changes in the geometric structure of the template are minimal. Mesh models created by template fitting are thus sometimes called homologous models. The advantage of template fitting is that the template can be deformed and fitted to different parts of a target object that are spatially close but distant from the surface (such as the zygomatic arch and temporal region of the skull) without affecting each other’s deformations. By analyzing coordinates of the vertices that make up the mesh models using a multivariate analysis method, such as PCA, variations in the entire surface shape can be analyzed, and virtual shapes at arbitrary positions in the distribution can be calculated and visualized. Mesh models created using template fitting have been widely used for shape analyses in various fields”.

Various new findings have been reported using this technique. Suzuki et al. [8] reported the method for sex identification using the mandible; this method has been reported to have an accuracy rate equal to or higher than that of conventional methods [8,10]. Inoue et al. demonstrated the effectiveness of this model in the 3D comparison of the ramus morphology between the contralateral and deviated sides in asymmetric mandibles [11]. Albouga et al., by comparing the pre- and post-operative images in orthognathic surgery, illustrated that the lower facial height was decreased, and the chin and lower lip moved posteriorly [12]. Yasuda et al. reported the difference between the posed smile and straight face. Although their results were clinically apparent, they reported that this article is the first to statistically verify the same [13]. In addition, Suzuki et al. reported that the main component values of a posed smile could be predicted from the main component values of a straight face using multiple regression analysis. They found the error of the PC method and conventional method, including the homologous modeling techniques and principal component analysis, were clinically small and useful for predicting change in facial expression [14]. The reports using homology modeling are increasing in the medical field. In these reports, all present virtual shapes, such as average shapes. The present study focused on this technology—visualizing virtual shapes using homologous models—applying it to predict the original shape of the defects of a mandible. Conventional reconstruction of a partial defect in a mandible involves predicting the shape of the fractured part based on the shape of the opposite side. However, this method is based on the misinterpretation that the mandible is symmetrical.

On the other hand, reconstruction of mandibular defects after trauma and tumor resection is one of the most challenging problems facing maxillofacial surgeons. Historically, various autografts and alloplastic materials have been used in the reconstruction of these types of defects. Hidalgo reported the utility of vascularized fibula flaps for mandibular reconstruction in 1989 [15]. Since then, the fibular free flap has become the first option for mandible reconstruction [16,17]. This flap has many advantages, including the high quality of the long bicortical bone grafts, long pedicle, and wide vessel, and the ability to incorporate skin and muscle, which are required for mandibular reconstruction [18,19,20]. However, it is difficult to achieve the ideal functional and esthetic outcomes because there is a fundamental difference between the morphology of the mandible and the fibula.

Currently, virtual surgical planning using CAD/CAM and 3D printing technology provides a valuable tool to support accurate surgical planning and precision in mandibular reconstruction [21]. In addition, 3D printing technologies have recently made it possible to arbitrarily shape metals such as titanium [22]. Advances in these techniques have made it very important to predict the shape of the mandibular defects. In this study, the error between the predicted and actual mandibles was as much as 1 mm. If we can reduce the CT slice width to less than 1 mm by improving the positioning of landmarks, we believe that this method will be applicable in actual clinical practice, potentially eliminating esthetic disorders in patients after jaw reconstruction.

## 5. Conclusions

Homologous modeling techniques are an innovative technique to assess morphology. We believe that this method will be applicable in actual clinical practice, potentially eliminating esthetic disorders in patients after jaw reconstruction.

## Figures and Tables

**Figure 1 jpm-12-01831-f001:**
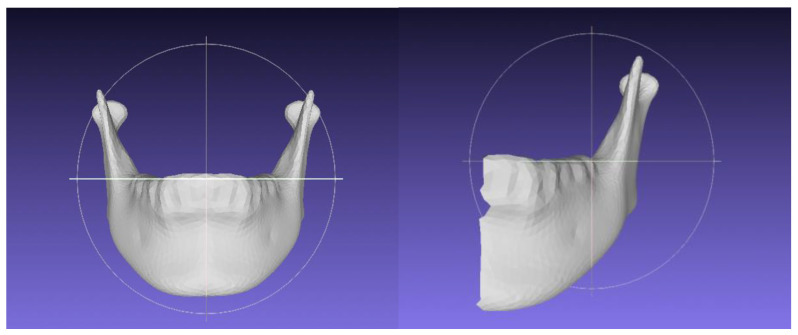
The homologis model of total mandible (**left side**) and half mandible (**right side**).

**Figure 2 jpm-12-01831-f002:**
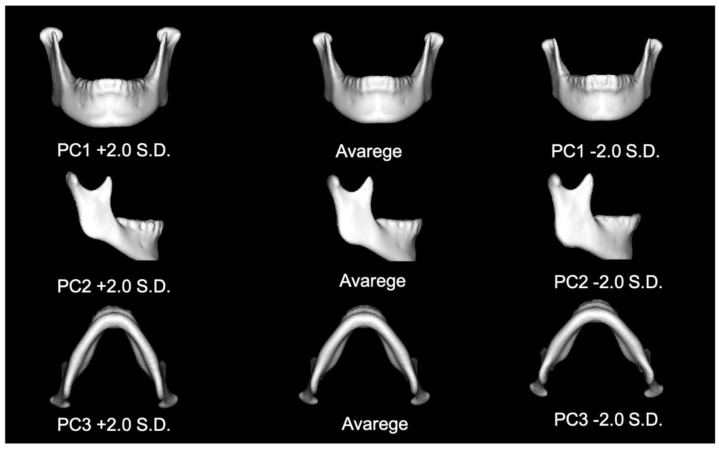
The figure of PC 1, 2 and 3 in total mandible.

**Figure 3 jpm-12-01831-f003:**
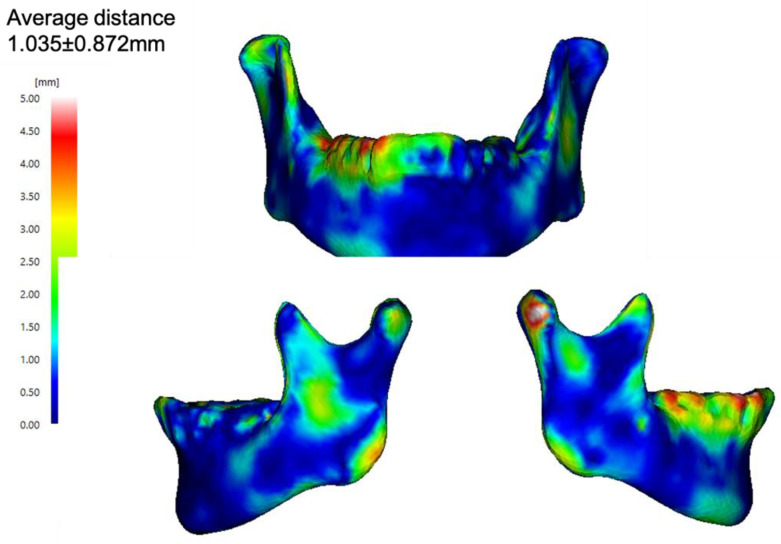
In face-to-face distance, the mean absolute error between the predicted mandible and the actual mandible.

**Table 1 jpm-12-01831-t001:** The anatomical landmarks in total mandible.

	Landmarks		Landmarks
1	Right condylar head	11	Right mandibular angle
2	Left condylar head	12	Left mandibular angle
3	Right mandibuar notch	13	Mandibular plane between right lower incisor and canine
4	Left mandibular notch	14	Mandibular plane between left lower incisor and canine
5	Right coronoid process	15	Between right lower central and lateral incisor
6	Left coronoid process	16	Between left lower central and lateral incisor
7	Between right lower first and second premolar	17	Centre of mentaris
8	Between left lower first and second premolar	18	Mental spines
9	Between right lower incisor and canine	19	Between No 18 and No 20
10	Between left lower incisor and canine	20	Between right and left lower central

**Table 2 jpm-12-01831-t002:** The anatomical landmarks in half mandible.

	Landmarks		Landmarks
1	Right condylar head	7	Mandibular plane between right lower incisor and canine
2	Right mandibular notch	8	Between right lower central and lateral incisor
3	Right coronoid process	9	Centre of mentaris
4	Between right lower first and second premolar	10	Mental spines
5	Between right lower incisor and canine	11	Between No 10 and No 12
6	Right mandibular angle	12	Left lower central

**Table 3 jpm-12-01831-t003:** Principal component in total mandible.

Principal Component	Eigen Value	Conribution Rate (%)	Cumulative Contribution Rate (%)
1	1763.1	27.9	
2	773.4	12.2	40.1
3	548.0	8.7	48.7
4	467.7	7.4	56.1
5	271.1	4.3	60.4
6	249.0	3.9	64.3
7	208.7	3.3	67.6
8	179.4	2.8	70.5
9	165.1	2.6	73.1
10	140.0	2.2	75.3
11	132.3	2.1	77.4
12	122.2	1.9	79.3
13	121.6	1.9	81.2
14	112.2	1.8	83.0
15	89.6	1.4	84.4
16	82.5	1.3	85.7

**Table 4 jpm-12-01831-t004:** Interpretations of distinguishable component in PCA between homologous mandible model in total mandible.

Principal Component	Interpretation
PC1	Mandibular size
PC2	The length and angle of mandibular ramus
PC3	Mandibular arch (V shape or U shape)

**Table 5 jpm-12-01831-t005:** Principal component in half mandible.

Principal Component	Eigen Value	Conribution Rate (%)	Cumulative Contribution Rate (%)
1	3485.5	27.6	
2	1366.6	10.8	38.4
3	1134.9	9.0	47.4
4	853.9	6.8	54.2
5	585.1	4.6	58.8
6	440.1	3.5	62.3
7	404.9	3.2	65.5
8	355.5	2.8	68.3
9	321.7	2.5	70.8
10	283.2	2.2	73.1
11	250.9	2.0	75.1
12	228.1	1.8	76.9
13	218.6	1.7	78.6
14	208.1	1.6	80.2
15	184.6	1.5	81.7
16	164.4	1.3	83.0

**Table 6 jpm-12-01831-t006:** Principal component in estimate PC.

PC	1	2	3	4	5	6	7	8	9	10	11	12	13	14	15	16
Total Mandible	17.7	−65.1	37.6	5.8	−23.6	24.1	−1.2	4.3	11.6	1.0	10.3	−11.7	7.9	3.0	−2.3	5.9
Half Mandible	−9.7	−48.7	42.5	−9.9	1.7	−3.0	9.7	15.3	−4.6	8.4	−5.8	−2.7	3.5	4.1	−4.5	−0.9
Estimate PC	11.8	−71.0	53.7	18.8	−11.0	10.4	−8.8	5.4	17.8	−10.7	2.1	−4.3	−3.6	2.5	3.0	2.0

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
