# Peer review of "Application of the Homologous Modeling Technique for Precision Medicine in the Field of Oral and Maxillofacial Surgery"

_jpm, 2022, doi:10.3390/jpm12111831_

Round 1
Reviewer 1 Report
First of all, I would like to congratulate the authors for choosing this topic. The aim of this study was to establish a new method for predicting the original shape of a mandibular defect site using the homologous modeling technique.
Secondly, I recommend some revisions to the manuscript.
Please see my suggestions below.
Introduction
· References must be added after every sentence. The entire Introduction contains a lot of information and this must be properly cited. The only references that are included in the Introduction are 1 to 4, all linked to Mochimaru et al.
Materials and methods
· Lines 70-72 – ”All experimental protocols were approved by the institutional ethics committee at Osaka Medical College, Osaka, Japan. Informed consent was obtained from all patients.” The information regarding the ethics approval should be moved to the beginning of this section.
· Lines 64 – ”This study used data from 44 patients (22 men, 22 women)”. This information should be moved to the Results section.
· Lines 74-75 – ”Homologous modeling followed the methodology outlined by Suzuki et al. [3] (Figure 1).” Reference number 3 does not belong to Suzuki et al. Furthermore, please describe the methodology here, as well, so that the reader better understands the study.
· Figure 1 must be moved in the proximity of the paragraph where it is referred to.
· Table 1 and Table 2 must be moved in the proximity of the paragraph where they are referred to.
· Some statistical testing should be performed.
· Sample size estimation should be performed.
Results
· Every Table and Figure must be moved in the proximity of the paragraph in which they are referred to.
· Develop the obtained results in text, not only in tables.
· The Results section should contain information regarding the initial number of patients, the excluded number of patients, and the final number of patients.
Discussion
· The limitations of this study are missing and must be added at the end of the section.
Conclusion
· This section is very short and must contain more information.
Best regards!
Author Response
Thank you for your careful review.
I changed that you pointed out.
Best Regards

Reviewer 2 Report
I think it is a scientifically competent article.
Author Response
Thank you for your careful review.
Best Regards
Reviewer 3 Report
The study is well written and exposed. The results are clearly presente and I agree that this technique could be useful for the precise prediction of the mandible difect sizes, allowing a better oral rehabilitation.
Author Response

(The authors gave the same response as above.)

Round 2
Reviewer 1 Report
The authors didn’t highlight the reviews they made to the text according to my comments. The answers should be provided point by point, for every comment made.
As seen from the modified text, some changes were made, but some comments were not addressed.
Please see my suggestions below:
· Lines 52-54 – Please add reference.
· Lines 55-68 – Please add reference.
· The initial number of patients, and the excluded number of patients is not reported, as suggested in my previous comment.
· The sample size estimation is missing.
· The limitations of this study are missing or are not clearly highlighted at the end of the Discussion section.
Best regards!